# Artificial Cavities in the Northern Campania Plain: Architectural Variability and Cataloging Challenge

Marco Vigliotti ⓘ, Maria Assunta Fabozzi, Carla Buffardi ⓘ and Daniela Ruberti *ⓘ

Engineering Department, Campania University "L. Vanvitelli", Via Roma 29, 81031 Aversa, CE, Italy; marco.vigliotti@unicampania.it (M.V.); mariaassunta.fabozzi@unicampania.it (M.A.F.); carla.buffardi@unicampania.it (C.B.)
* Correspondence: daniela.ruberti@unicampania.it

**Abstract:** In Campania (southern Italy), the widespread presence of anthropogenic cavities in the subsoil of the Neapolitan and Caserta provinces is well known. In these towns, the underground quarrying activities were performed for centuries to extract volcanic tuffs for buildings. The urban developments have sealed many signals of the presence of cavities and their real extent is almost unknown, thus representing a geological hazard and contributing to the subsoil instability of many places. This contribution will show the main cavity typologies recognized across an area north of Naples according to the geological characteristics of the subsoil. The main aim of the study is the cataloging of the cavities and the analysis of the city subsoil as their presence may easily trigger the collapse of the shallow or deeper soils. Moreover, the recognition and sustainable reuse of cavities contributes to enhancing the cultural and touristic promotion of a territory. In this study, a database framework was elaborated that includes all the possible architectural, geological and geotechnical elements of the cavities. Data were managed in a GIS environment in order to provide a useful tool for monitoring and managing the cavities for risk mitigation and tourism enhancement.

**Keywords:** Campania Plain; Italy; Campania Grey Tuff; underground cavities; historical heritage; database

## 1. Introduction

Sinkhole (s.l.) phenomena are present all over the world and the attention that scientists, at an international level, are dedicating to the problem of their prevention is increasingly growing [1–5]. This type of instability may have natural origins, mostly related, for example, to collapse of karst cavities, suffosion processes and siphoning [3,5–8]. Nevertheless, in many urban historical centers, this phenomenon can be triggered by the occurrence of underground anthropic cavities [5,9–11]. The latter represent, in fact, a great threat to the safety of the urbanized areas due to their relative unpredictability. The need to carry out a survey of underground quarrings in urban centers has two reasons:

(a) The anthropic cavities represent an absolute documentary value, still unduly neglected and little used for the purposes of a correct and sustainable management of the territory, natural resources and historical and artistic heritage. The enhancement and sustainable reuse of cavities contributes to enhancing the cultural and touristic promotion of a territory.

(b) In correct urban management, the knowledge of the city subsoil is a priority as the presence of cavities may easily trigger the collapse of the shallow soils or deeper subsoil [9,12–16].

The contribution of the analysis of documentary sources, including oral ones, and of historical cartography is fundamental in attempting to reconstruct the presence and extension of cavities in an urban environment. The result is a large mass of data which are, however, extremely heterogeneous, fragmented and dispersed.

Since 1928, the awareness of the large diffusion of natural cavities throughout the Italian territory has pushed the creation of a classification provided by the National Commission on Artificial Cavities of the Italian Speleological Society (ISS; [17]) and later adopted by the International Union of Speleology (IUS; [18]) based on data that define the topographic, speleometric and unique identification of each cave; these data are now available at https://speleo.it/catastogrotte/ (accessed on 19 April 2023). Over the years, the recurrence of sinkhole phenomena in urban areas has highlighted the need to know the subsoil architecture of inhabited centers [5]; therefore, interest has also focused on cavities of anthropic origin and, over time, on creating a Classification of Artificial Cavities organized in a very similar way to that of the Natural Caves [18].

The catalog of the cavities becomes a fundamental tool for the knowledge and protection of an underground heritage; therefore, the design, construction and populating of a geodatabase is of fundamental importance as a reliable and secure tool for archiving and preserving large quantities of spatially related data. Proper data organization makes the geodatabase a useful tool for analyzing the data to obtain important information to support urban management and, in particular, to relate cavities with the topsoil (buildings and utilities).

In Campania (southern Italy), sinkhole phenomena induced by the widespread presence of anthropogenic cavities in the Neapolitan and Caserta provinces are frequent and well known [14,19–23]. Nevertheless, in many urban centers of this area, cavities have been reported in specific geological investigations, although their real extent is almost unknown. In these towns, the underground quarrying activities were performed to extract volcanic tuffs for buildings. The urban developments have sealed every signal of the presence of cavities, which thus represent a geological hazard and contribute to the subsoil instability of many places [8,10,24–29].

This study reports a preliminary investigation carried out in the Campania Plain north of Naples, an area which is characterized by the high amount of volcanic tuffs in the subsoil. Tuffs and cinerites are the main lithologies recognized in the first tens of meters. Since the tuffs have good mechanical properties (i.e., high compression strength, low specific weight), they have been subject to extensive quarrying activity since antiquity [30].

The main goals of the paper are to: (i) highlight the relation between the geological framework and anthropic activities related to urban subsoil; (ii) highlight the main extraction characteristics (architecture); and (iii) propose a cataloging model to relate to the National Geodatabase of Artificial Cavities proposed by the ISS to support the management of hazards related to the quarrying activities.

## 2. Study Area

The study area in the northwestern part of the Campania region (Italy; Figure 1) corresponds to the northern and northeastern part of the Campania Plain, a broad, complex graben closely controlled by NE–SW, NW–SE and E–W normal fault activity and established in the Late Pliocene [31] or Early Pleistocene [32,33] era along the Tyrrhenian side of the Apennine Mountains. The sedimentary evolution of this graben was conditioned by the fluvial and marine processes and volcanic activity of the Campi Flegrei, Somma-Vesuvious and Roccamonfina volcanoes [34–38].

The considered sector is characterized by a flat morphology between 95 and 20 m a.s.l. In this area, the subsoil is formed by the succession of different units composed of volcanoclastic deposits, particularly in relation to the Campania Grey Tuff (CGT; 39 ky B.P; [39]) and Neapolitan Yellow Tuff (NYT; 15 ky B.P.; [40]) pyroclastic eruptions from the Campi Flegrei volcanic district [16].

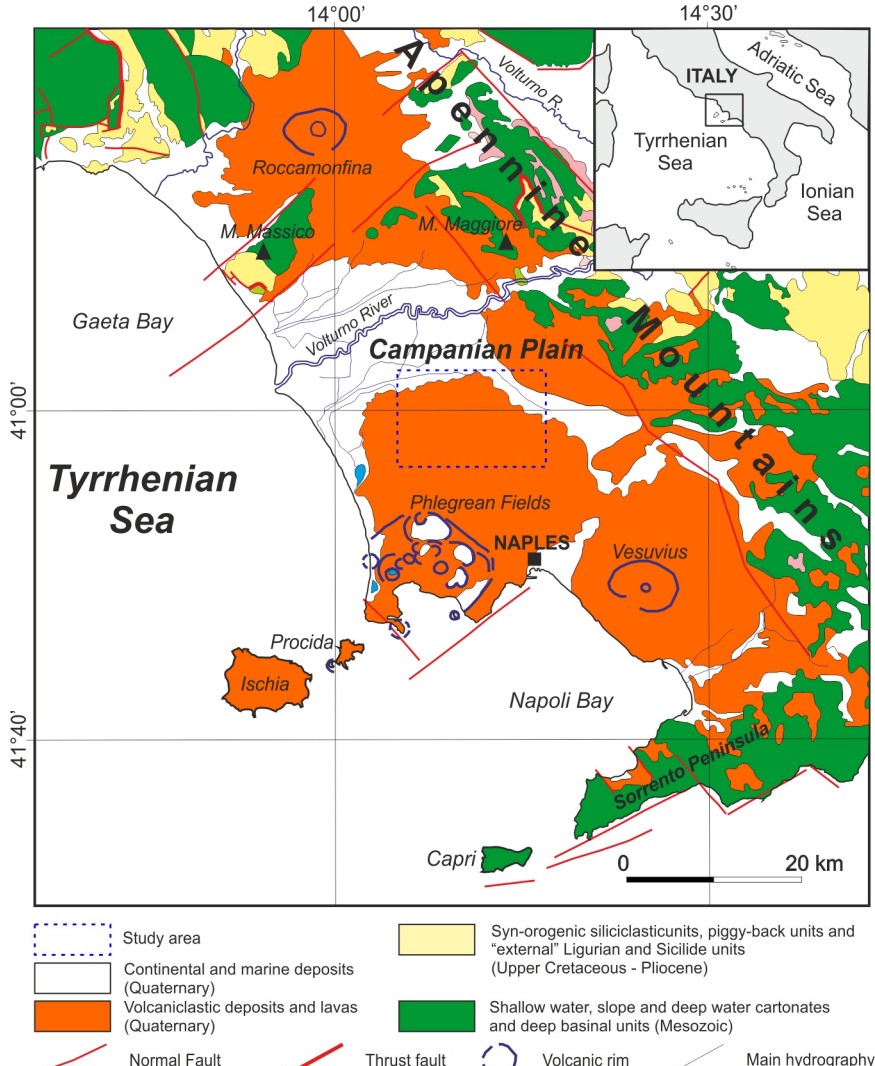

**Figure 1.** Geographical location and simplified geological map of the northwestern part of the Campania region.

The CGT deposits, in particular, settled over the whole Campania Plain, giving rise to a thick (up to 40 m thick), laterally continuous volcanoclastic unit. Different lithofacies can be recognized within the CGT, which are mostly derived from the different mineralogic composition [41–43]. On the whole, from top to bottom, the succession of CGT lithofacies is characterized by: an upper incoherent part represented by the cinerazzo (cz); the coherent zeolitic yellow tuff (tgz; LYT—Lithified Yellow Tuff—in Cappelletti et al., 2003 [42]) and/or grey tuff (tg; WGI—Welded Grey Ignimbrite—in Cappelletti et al., 2003 [42]), piperno tuff (tp), and pipernoide (pp) tuff; and cinerite (cn), soft lithofacies which occur close to the bottom of this unit. These lithofacies greatly vary in thickness and occurrence vertically and laterally (towards the east or north) from the source area. The uppermost unit of the reconstructed stratigraphic succession is represented by the thin, grey and loose ashy deposits of the NYT, which are locally separated from underlying CGT by a paleosol [16,30,44,45].

The good mechanical characteristics of the tuff lithofacies (Table 1) justify the presence of numerous quarries and/or cavities according to the availability of adequate thicknesses of coherent lithofacies (tgz, tg; [15]).

**Table 1.** Average geomechanical features of the tg and tgz lithofacies (from Langella et al., 2013 [46]).

| | Dry Density (kN/m³) | Specific Gravity (kN/m³) | Open Porosity (%) | Imbibition Coefficient (%) | Compressive Strength (UCS) (MPa) |
|---|---|---|---|---|---|
| Grey facies (tg) | 11.42 | 25.53 | 55.24 | 41.61 | 5.23 |
| Yellow facies (tgz) | 10.97 | 22.68 | 51.61 | 34.73 | 6.45 |

## 3. Materials and Methods

Geological data derive from the study of about 210 lithostratigraphic logs from boreholes reaching a 40 m maximum depth, available from different bibliographic sources. Some additional subsurface information was obtained from excavation for constructions (in the northern part of the study area), exposing about 10 m of deposits.

The stratigraphic marker to correlate the different units was represented by the CGT volcaniclastic deposits.. The lithological descriptions were interpreted and homogenized in terms of lithologic units and collected into a relational geodatabase which also contains the related thickness and upper- and lower-boundary elevations a.s.l.

Previous studies [15] and some urban management plans have highlighted the presence of numerous cavities in the area. A geodatabase was elaborated to organize previous data and the results of new, specific investigations, which also considers the position and planimetric development of the cavities.

The first step (requirement analysis) concerned the identification of the data to be managed and the relationships between them; the conceptual design step defined the entities and the relationships between them; the logical design step represented the final phase of database construction through the identification of the tables and the relationships between them (Figure 2).

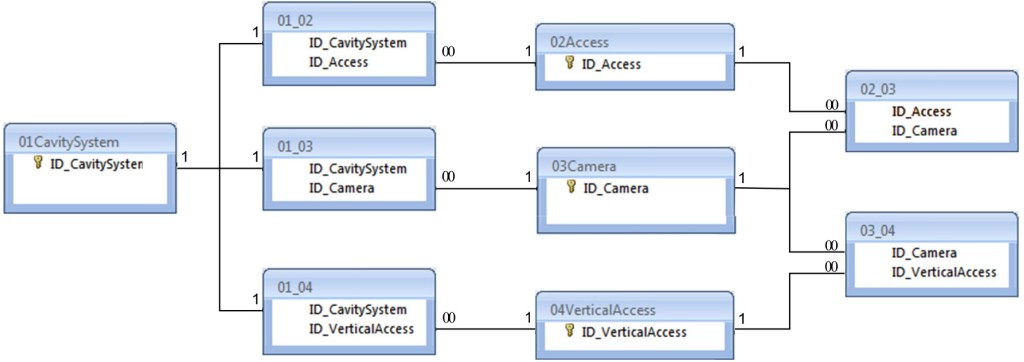

**Figure 2.** Structure of the database created for cavity system management showing the relationships ("one-to-one" and "one-to-many") between the tables of data.

Where available, the plans of the cavities were (i) scanned and (ii) georeferenced by identifying the GCP (Ground Control Point) and were then (iii) vectorized. All collected data are managed through the GIS project based on a regional technical map in 1:5000 scale.

## 4. Results and Discussion

### 4.1. Geological Setting

Geological data analysis has revealed common features for the considered area (with minor differences linked to the different thickness of the various lithofacies of CGT):

- The occurrence of coherent zeolitic yellow tuff was the main trigger for cavity development. The mean thickness of 8 m made this tuff excellent for quarrying purposes. The zeolitized yellow welded tuff is formed by an ashy matrix with rounded lapilli and dispersed pumice clasts; few and scattered coarse scoriae are present. This material is good for making bricks for walls and buildings.

- Below the zeolitic yellow tuff, slightly welded pyroclastic soils and ash layers occur showing different compaction degrees. Their occurrence conditioned the depth of extraction.
- In the slightly lithified upper part, the volcanic ash is characterized by abundant pebbles/breccias of volcanic tuff (cappellaccio or pebble tuff) and represents the soft deposits through which the tuff easily passes.
- A one-meter thick paleosoil underlines the upper limit of the CGT. The former is formed by brown, silty-sand, pyroclastic units with small rounded and altered pumices; this could represent an impermeable barrier to rainwater and water infiltration.

The upper deposits are mainly composed of white-yellowish and greyish cinerite in which pumice layers intercalate, mostly related to more recent eruptions of the Phlegrean Fields. Towards the east, the thickness is reduced to a few meters (about 2 m).

Taking into account the three-dimensional distribution of the CGT lithofacies and depth, it follows that (Figure 3):

- The depth of the CGT top is higher in the southern part of the study area (Parete, 47.50 m a.s.l.; Sant'Arpino, 31.36 m a.s.l.) and lower in the northern part (Orta di Atella, 10.40 m a.s.l.; Frignano, 11.20 m a.s.l.); accordingly, the depth of cavity development is also different.
- The topographic surface reflects the trend of the tuff top, according to Ruberti et al. (2020).
- *tg*z thickness ranges between 1.3 and 13.2 m; this lithological unit is heteropic with tg towards the south and east and lies above dense coarse sands (cinerazzo).
- The post-CGT deposit thickness (about 8.5 m on average) is not uniform across the area, ranging from 17 m (in the southeastern side) and 5 m (to the north, close to the Regi Lagni Canal).

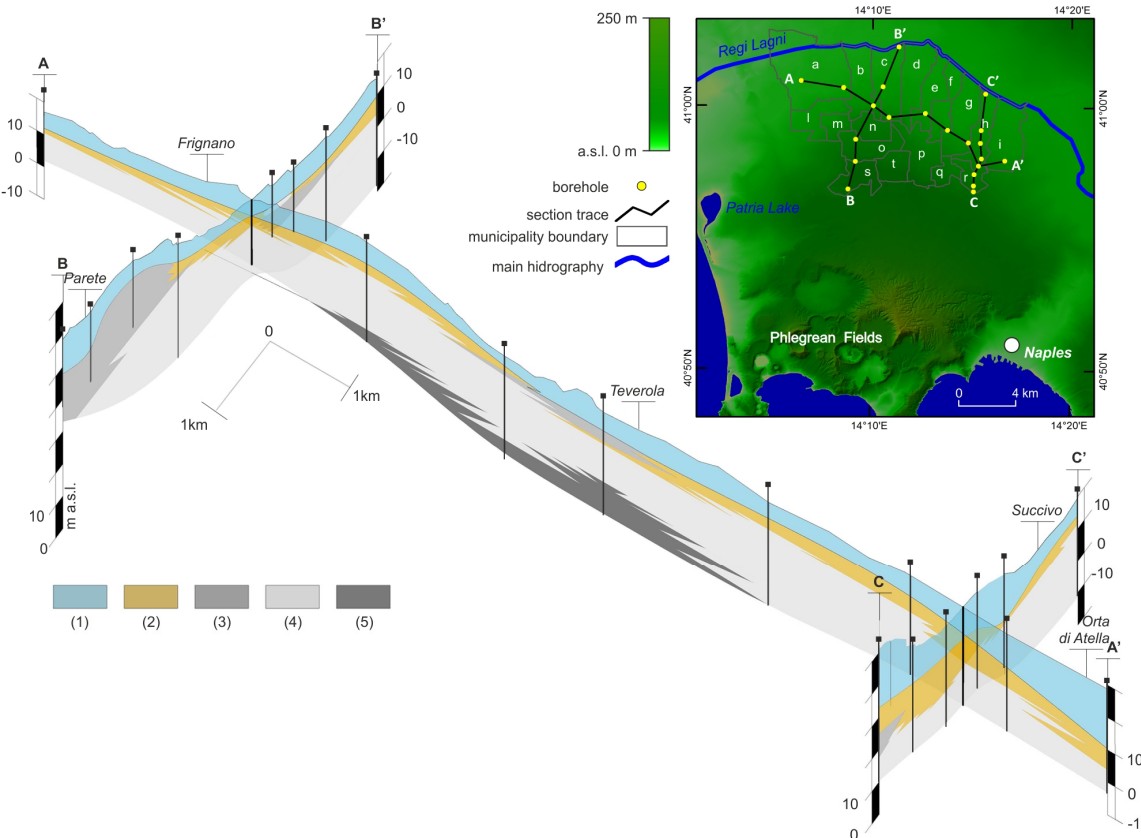

**Figure 3.** Fence diagram of geological cross sections illustrating the three-dimensional distribution of the recognized Campanian Ignimbrite facies: (1) post-CGT, (2) zeolitic yellow tuff, (3) gray tuff, (4) "cinerazzo",

(5) piperno; (a) Casal di Principe, (b) Villa di Briano, (c) Frignano, (d) Casaluce, (e) Teverola, (f) Carinaro, (g) Grigignao d'Aversa, (h) Succivo, (i) Orta di Atella, (l) San Cipriano d'Aversa, (m) Casapesenna, (n) San Marcellino; (o) Trentola—Ducenta, (p) Aversa, (q) Cesa, (r) Sant'Arpino, (s) Parete, (t) Lusciano. DEM based on TINITALY/01 data [47].

Knowledge of the geological and stratigraphic architecture of subsoil gives a great advantage in forecasting the presence of cavities and their depth where they are unknown.

### 4.2. Architecture of the Cavities

On the whole, the development of cavities and quarries was strongly linked to the lithology of the CGT and the related thickness (Figure 4). A dense cavity network characterizes most of the historical center of the towns developed within the yellow tuff containing large scoriae, pumices and lava blocks. Pipe structures are common. They represent the site of degassing ash flows, vary from centimeters to meters in width and are characterized by fines depletion and weak, loosely packed deposits that allowed the quarring activity to start.

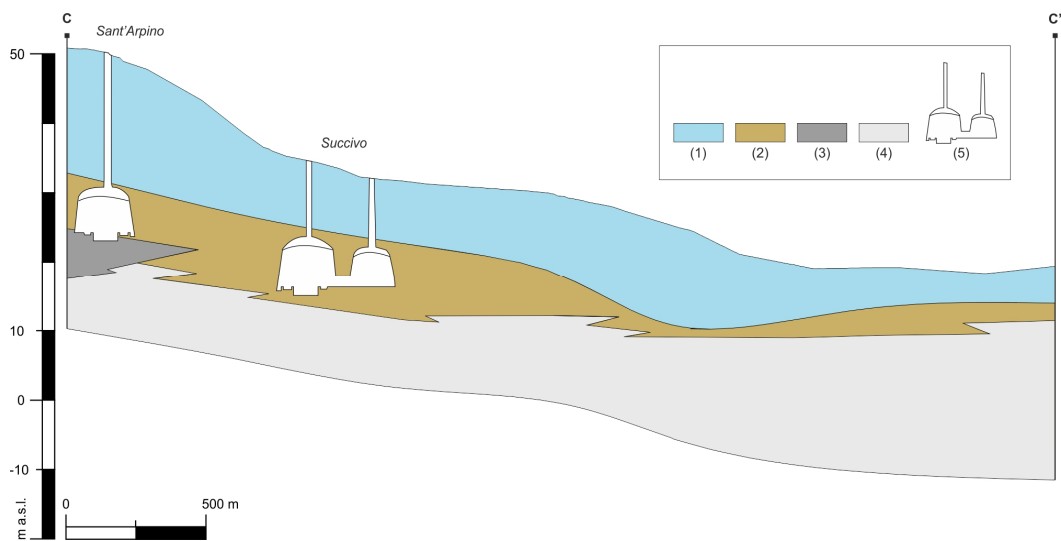

**Figure 4.** Geological cross-sections showing the relationship between cavities and CGT lithofacies (track C-C′ in Figure 2): (1) post-CGT, (2) zeolitized yellow tuff, (3) gray tuff, (4) "cinerazzo", (5) cavities. Not to scale.

The creation of a cavity system (Figure 5) initially began with a vertical circular or squared excavation (never less than 1.5 m wide—known locally as "occhi di monte") made from the ground surface into the upper loose pyroclastic deposits via the underlying tuff unit. The excavation of the cavity was carried out as a "bottle" or a "bell". The depth of the tuff roof is rarely less than 1 m.

The vertical access points were sometimes covered by retaining walls made up of tuff bricks resting on the tuffaceous bank; in correspondence with the access well, sometimes the excavation could continue in depth starting from the base/floor of the cavity until it reached the groundwater. These latter shafts (known locally as "canne di pozzo") usually have a square section (about 1 × 1 m) and small furrows dug along the entire length as a kind of stairs to allow inspection.

A single excavation could be joined to others at a certain distance in such a way as to determine the coalescence of several chambers through the creation of narrow tunnels or wide "segmental arch" passages. Generally, each single chamber has a sub-rectangular planview with a vault shape of "trapezoid-shaped arch" or "segmental arch" (Figure 6). Adjoining rooms can be separated by rock pillars encased in the tuff. Therefore, the final planimetric development of the cavity system resulted in a complex network of chambers and passages.

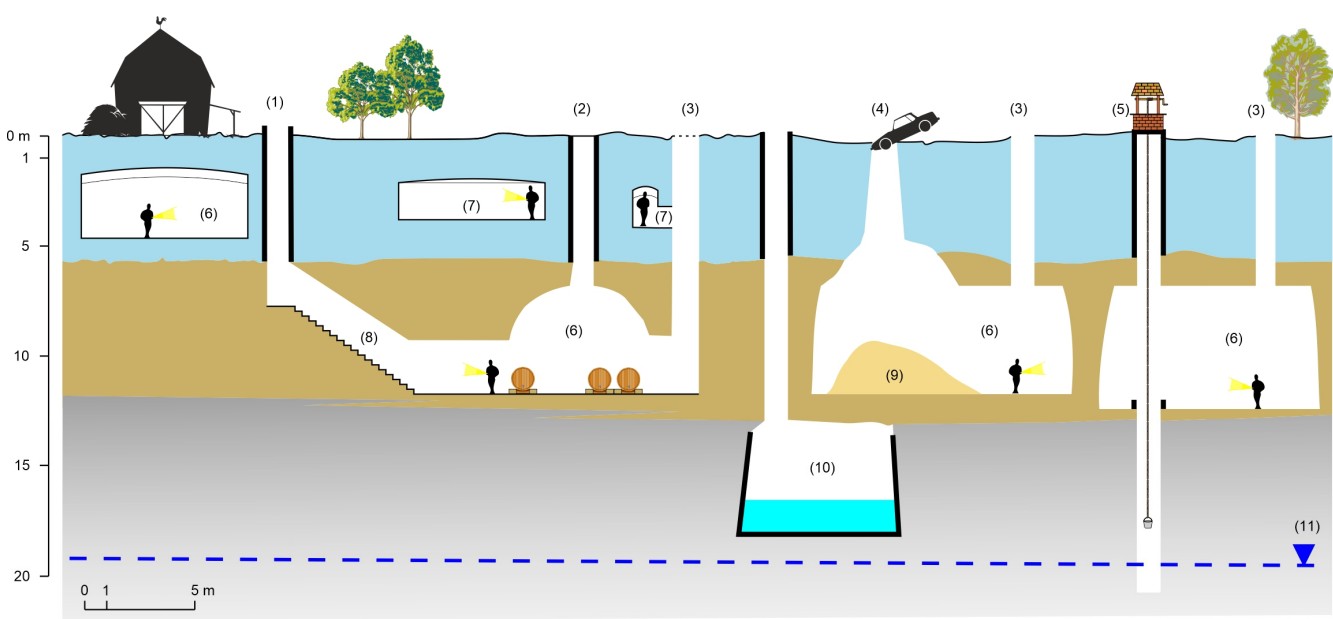

**Figure 5.** Sketch of a cavity system: (1) vertical access covered by retaining walls, (2) buried vertical access, (3) vertical access well (4) sinkhole, (5) well for water extraction, (6) chamber, (7) "tana di lapillo", (8) stairs, (9) collapsed material, (10) water cistern, (11) water level. Lithology key as in Figure 3.

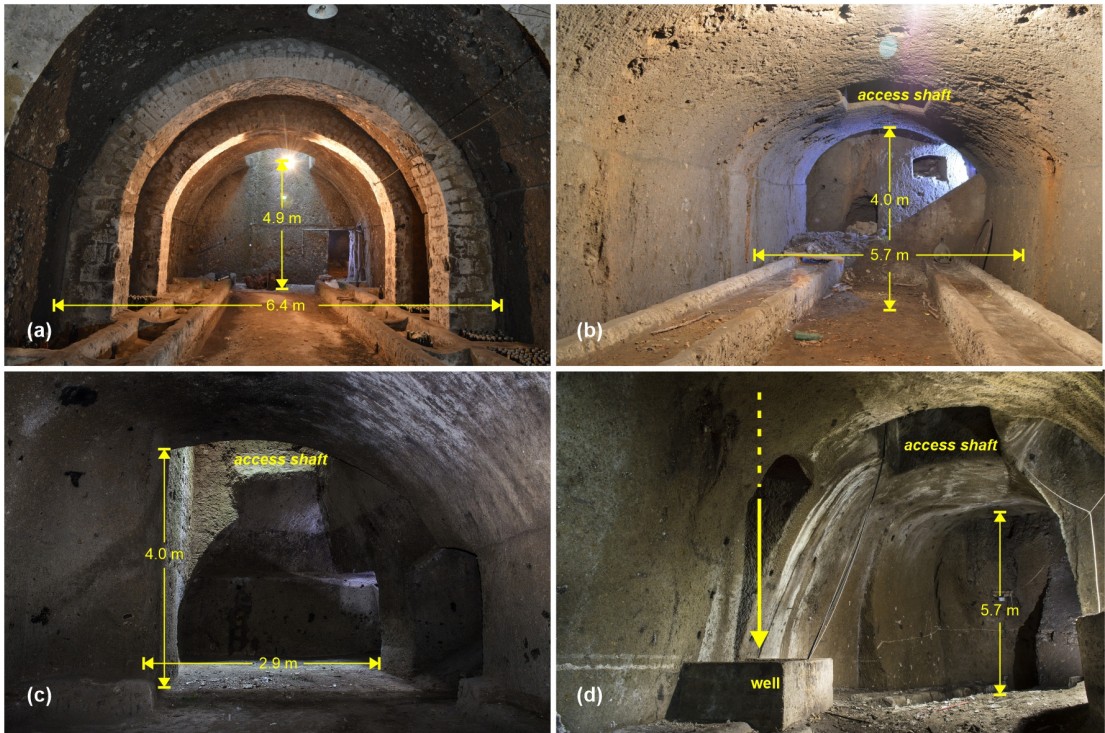

**Figure 6.** Some architectural types of arches of the vaults: (**a**) semicircular arch; (**b**) segmental arch; (**c**) rampant arch; (**d**) trapezoid-shaped arch.

When the overall thickness of the lithoid tuff allowed it, chambers were dug even deeper; they were plastered with a mortar produced using loose volcanic products ("pozzolane", *pulvis puteolanus*; [48,49]), which ensured internal waterproofing. In this way, the chamber constituted a cistern, a water storage system of specially conveyed rainwater which was used for domestic and agricultural purposes.

To use the cavities, accesses were created (represented by slides, a structure with a smooth sloping surface (Figure 7a) or by a system of stairs (Figure 7b) with one or more ramps), with steps directly carved into the tuff or made with bricks of the same material; the vault usually has a "round arch" shape. Sometimes, the access passages are wide, with a typical "segmental arch" vault, and consist of steps in the middle and at the lateral slides (Figure 7c).

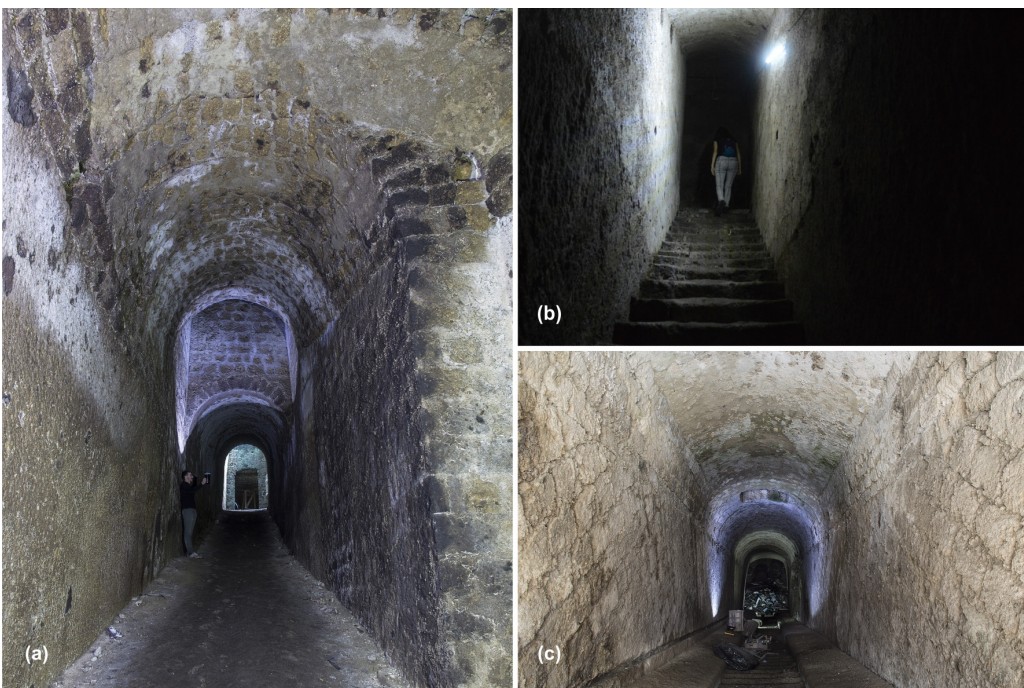

**Figure 7.** Access type: (**a**) incline ramp, (**b**) "L"-shaped stairs, (**c**) straight stairs with slides (1.5 m wide).

Cavity systems are also found in the deposits above the tuff bank, sometimes communicating/interconnected with the deeper ones. They generally have a rectangular planview and a "lowered arch" vault and are covered with tuff ashes or plastered with pozzolana-based mortars.

A minor, less-known cavity system consists of narrow sub-horizontal tunnels (known locally as "tane di lapillo") where the volcanic deposits are rich in pumice or lapilli and is used for construction purposes [48,49].

Uses of Cavities

The survey of the cavities and their uses over time in the area considered has, first of all, allowed to catalog them in the categories identified in the Classification of the Artificial Cavities (after [18]). The main types of underground cavities refer to type E.1 of the aforementioned Classification, i.e., "quarries" for building materials [15]. These are quarries for the extraction of tuff used as a building material and of pumice and lapilli used for making mortars; traces of manually conducted excavations are evident on the walls of the chambers, such as narrow grooves made with chisels and wedges to isolate large blocks of rock. When the extraction activity was terminated, the cavities were then used as "cisterns" to store rainwater (category A.4; Figure 8a) or "cellars" for storing foodstuffs (Figure 8b) and wine (category B.4; Figure 8c).

The latter use is still very common (Figure 8d) as the physical characteristics of the tuff favor the fermentation of the musts, ensuring temperature values (average temperature values = 12 °C) and humidity (average humidity values = 61%) constant in underground environments, which is also guaranteed by the access drillings and by the wells which function as ventilation shafts. Of particular interest is the production of a vine typical only

of this area, the asprinio, which dates back thousands of years and whose production has become so important in the last decade as to require recognition of the area as a UNESCO World Heritage Site. This has led to an increase in the production of asprinio wine and the reuse of some abandoned hypogea for this purpose.

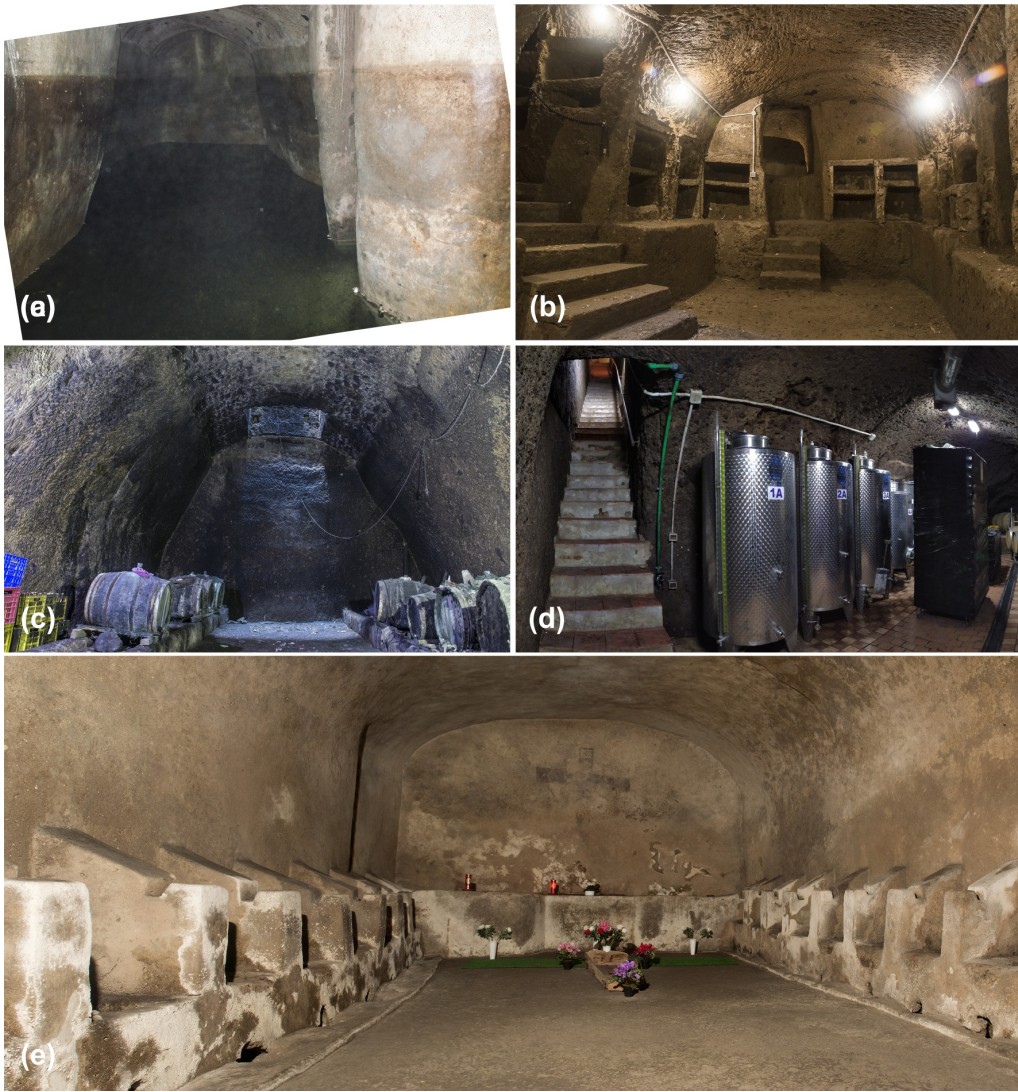

**Figure 8.** Some examples of uses of artificial cavities: (**a**) water cistern (historical center of Aversa)—note the waterproof plaster along the tuff walls to prevent water from infiltrating the walls; (**b**) well-preserved cellars quarried below a worship place in the historical center of Aversa; (**c**) abandoned cellar used for wine production—note the old wooden barrels musty from humidity (Sant'Arpino); (**d**) cellar still used for vine production (San Marcellino); (**e**) burial place below a cloister in a monumental worship complex (Aversa).

Many cavities, on the other hand, were used during the Second World War as "war shelters for civilians" during air raids (category D.7), especially in historic centers whose houses were built with the extraction of tuff from the subsoil. Finally, below the main monastic complexes, the cavities were destined to become "burial places" and/or "chapels" (category C.1 and C.2; Figure 8e).

Currently, less than 1% of the cavities are used as wine cellars; the remainder is abandoned and/or lost. Moreover, the growth of the built-up areas has often caused knowledge of where the cavities were located to be lost. Once abandoned, these cavities may undergo degradation of the tuff mass if not properly maintained, owing to weathering

processes and water infiltration that significantly affect the physical properties of the rock (cf. [11]). One of the main triggers for susceptibility to sinkholes, in particular, derives from the excavation wells, especially the hidden ones; the saturation of sandy-loam soils due to water leaks from underground users reduces the shear strength of the loose ash deposits and induces collapse phenomena (Figure 9).

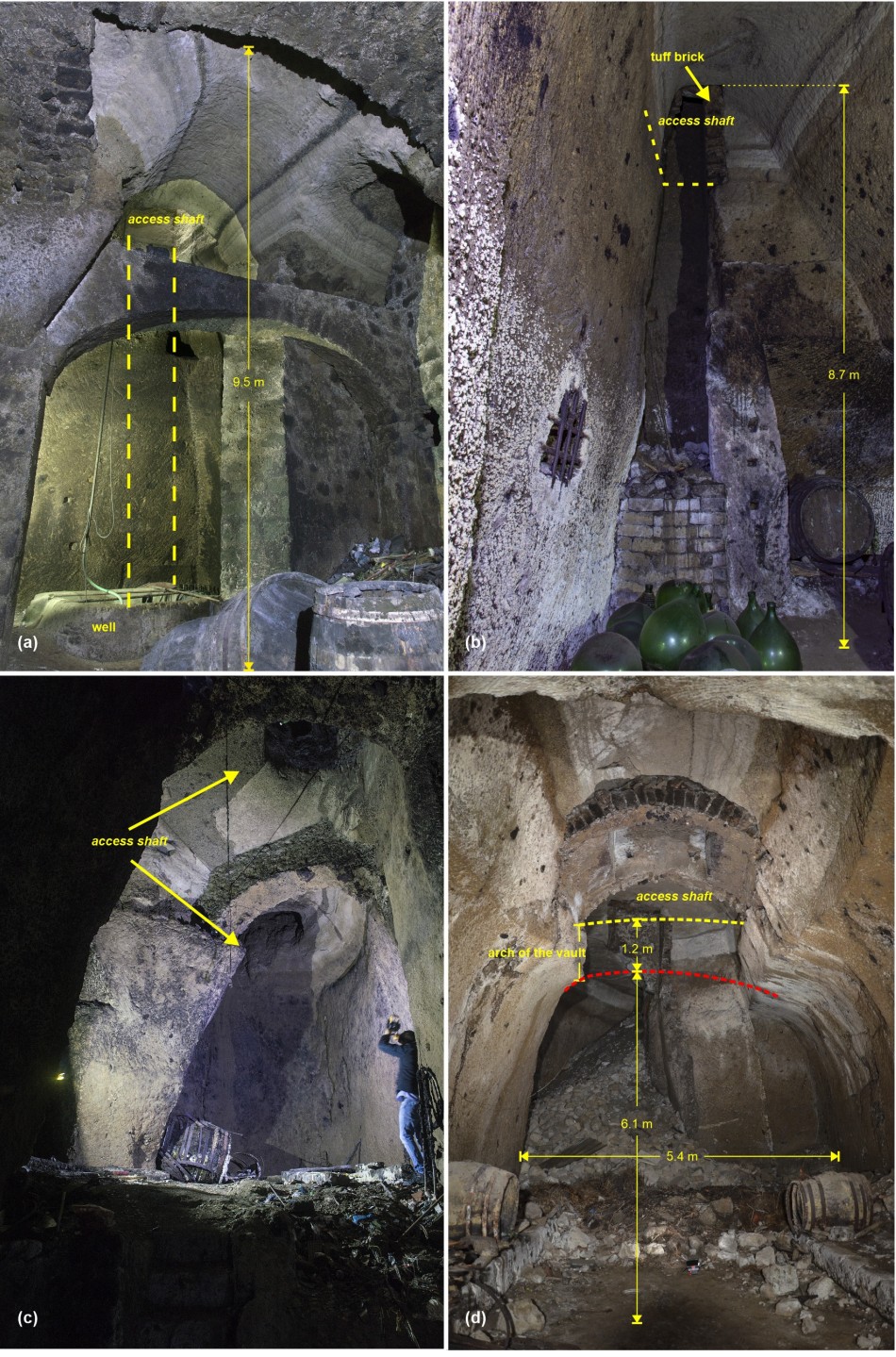

**Figure 9.** Collapses in proximity to the vertical well involving the loose pyroclastic ashes above the tuff and, locally, also the tuff roof. Large portions of the roof have collapsed either where the internal walls of the wells were covered with tuff bricks (e.g. San Marcellino, (**b**)), Succivo, (**d**) or where they did not (e.g. Orta di Atella, (**a**)), Sant'Arpino, (**c**). In some cases, masonry support works were built after the collapses (as in (**a,b**)).

### 4.3. Catalog of the Cavities

The survey carried out is far from exhaustive because about two thirds of the underground world is still unknown. However, it has highlighted the widespread diffusion of the cavity system in the subsoil of the urban areas of the considered territory, the great heterogeneity of the documentary reports, and the architectural variability. The need therefore arises to homogenize the information acquired and to catalog it in order to be able to analyze and manage it. It was therefore necessary to create a relational geodatabase that takes into account the peculiarities of local realities and provides for the collection of data of each part (chambers, wells, descents) of the cavity system. The one proposed in the present work represents an adaptation and an implementation of the previously mentioned National Geodatabase of Artificial Cavities, which defines a basic level of knowledge [18].

Based on the conceptual scheme shown in Figure 2, the first knowledge base level (Figure 10a) provides a table in which data about the plano-altimetric location, the volume, the lithology extracted, the characteristics of the cavity system, the past and current uses and the state of conservation are entered. A second level of detail provides tables in which the data of each part of the cavity system (access, Figure 10b; room, Figure 10c; wells, Figure 10d) are stored if detected. Here, in addition to the dimensional data, the architectural characteristics are also explained. The database is implemented with a photographic report, plans, sections and 3D models for each hypogeum.

**CAVITY SYSTEM**

| name field | data type | note |
|---|---|---|
| ID_CavitySystem | text | Primary key |
| PreviousKnowledge | yes/no | |
| ID_CavitySystemSource | text | |
| Note1 | text | |
| Detected | yes/no | |
| DateOfDetection | date | |
| Town | text | |
| Province | text | |
| Address | text | |
| Owner | text | |
| Easting | number | m |
| Northing | number | m |
| TopographicAltitude | number | m |
| Depth | number | m |
| LinearExtension | number | m |
| Area | number | m$^2$ |
| Volume | number | m$^3$ |
| Lithology | text | |
| EpochOfRealization | text | |
| YearOfRealization | number | |
| NumberOfAccesses | number | |
| NumberOfVerticalAccesses | number | |
| NumberOfRooms | number | |
| Type | text | unique space, two or more spaces |
| PlanimetricShape | combo | rectangular, trapezoidal, apsidal, cross, L-shaped, T-shaped |
| ConnectionBetweenRooms | combo | tunnel, passage, hallway |
| OtherConnectionsLikely | yes/no | |
| TypeConnections | text | |
| ClosedBy | combo | bricklaying, filling, collapse/landslide |
| Practicability | combo | simple, hard |
| PracticabilityNote | text | |
| HistoricalInterest | yes/no | |
| HistoricalInterestNote | text | |
| Classification | combo | based on Parise et al., 2013 |
| SecondaryCavealSystem | yes/no | |
| SecondaryCavealSystemNote | text | |
| GeneralStateOfConservation | combo | optimal, good, mediocre, bad |
| Note2 | text | |

(a)

**CAMERA**

| name field | data type | note |
|---|---|---|
| ID_Camera | text | Primary key |
| ID_CavitySystem | text | |
| DepthFromTheFloor | number | m |
| PlanimetricShape | combo | rectangular, trapezoidal, apsidal |
| ShapeRegular | yes/no | |
| ArchType | combo | flat arch, semi-circular arch, segmental arch, acute arch, rampant arch |
| Covered | yes/no | |
| CoveredFrom | combo | plaster on the wall, plaster on the celling, masonry on the wall, masonry on the ceiling |
| Length | number | m |
| Width | number | m |
| Height | number | m |
| ArchHeight | number | m |
| WallHeight | number | m |
| Access | yes/no | |
| NumberOfAccesses | number | |
| VerticalAccess | yes/no | |
| Pipes | yes/no | |
| Instability | yes/no | |
| CoolingJointsInTheArch | yes/no | |
| CoolingJointsInTheWall | yes/no | |
| CoolingJointsInThePillars | yes/no | |
| CoolingJointsInTheFloor | yes/no | |
| CrackInTheArch | yes/no | |
| CrackInTheWall | yes/no | |
| CrackInThePillars | yes/no | |
| PresenceOfWater | yes/no | |
| PresenceOfWaterType | combo | drip, flow, flooding |
| Pillars | yes/no | |
| MasonryArches | yes/no | |
| Well | yes/no | |
| Shelves | yes/no | |
| SubsequentAmendments | yes/no | |
| InUse | yes/no | |
| Cellar | yes/no | |
| StoregeArea | yes/no | |
| Tank | yes/no | |
| Garage | yes/no | |
| Other | yes/no | |
| OtherType | yes/no | |
| Note | yes/no | |

(b)

**ACCESS**

| name field | data type | note |
|---|---|---|
| ID_Access | text | Primary key |
| ID_CavitySystem | text | |
| Type of access1 | combo | public in an open space, public building in an open space, private in an open space, private building |
| Type of access2 | combo | ground level, shaft, incline, incline with ramps, straight stairs, straight stairs with slides, "L" shaped stairs, stairs with more ramps, spiral stairs, complex stairs |
| StepsMadeFrom | combo | tuff ashlar, lava stone, cement, mixed |
| Covered | yes/no | |
| CoveredWith | combo | plaster on the wall, plaster on the celling, masonry on the wall, masonry on the ceiling |
| Shelves | yes/no | |
| Side niches | yes/no | |

(c)

**VERTICAL ACCESS**

| name field | data type | note |
|---|---|---|
| ID_VerticalAccess | text | Primary key |
| ID_CavitySystem | text | |
| ID_Access | text | |
| ID_Camera | text | |
| VisibleOnTheSurface | yes/no | |
| Easting | number | m |
| Northing | number | m |
| CircularSection | yes/no | |
| Diameter | number | m |
| SquareSection | yes/no | |
| Side | number | m |
| RectangularSection | yes/no | |
| Length | number | m |
| Width | number | m |
| Covered | yes/no | |
| Notes | text | |

(d)

**Figure 10.** Database tables to manage data of the architectural elements of a cavity system: (**a**) general information of the cavity system, (**b**) chamber characteristics, (**c**) access point, (**d**) vertical access point.

*4.4. Spatial Distribution of the Cavities*

In the study area, the underground extraction of tuff certainly began in the late Middle Ages [50], and the age of extraction is closely linked to the age of the building under which the individual cavities are created since brick tuffs of irregular shape and size were used for construction. The quarring of tuff has undergone a significant decline since the 1950s [50] due to the diffusion of cement, the difficulty of finding skilled labor and the increase in production costs and manpower. Despite the fact that these activities had little impact from an environmental and landscape point of view, over time it was decided to use the open-air quarries of the Phlegrean hills and flat plain of the Caserta area, where the use of modern technologies allowed the rapid production of regular tuff bricks.

The lack of knowledge of the distribution of cavities in the Campania Plain is due to several factors. The extraction activity carried out underground, unlike that conducted "in open pits", is less visible and difficult to detect. This has led to insufficient historical and archival documentation of the cavities, which has contributed to their progressive disappearance from collective memory.

In addition to the lack of documentation, poor maintenance and conservation and the difficulty of accessing the sites have led to their inevitable oblivion over time.

Cavities are often hidden or protected by inaccessible structures; then, on the surface, the presence of cavities can be deduced by identifying wells usually protected by iron grates, cracks in the lava stone flooring the internal courtyards of ancient buildings or in gardens near water wells (Figure 11).

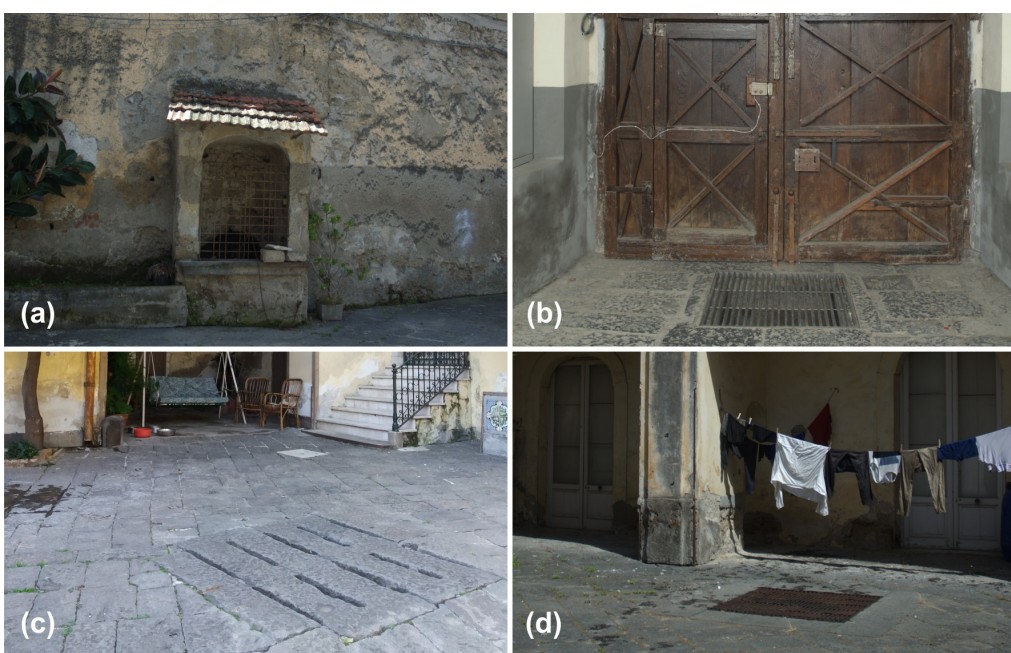

**Figure 11.** Indirect evidence of the presence of underground cavities: (**a**) buried well; (**b**–**d**) large iron and concrete grates (1 m wide on each side).

An indirect indication of the presence of underground cavities can be provided by the Topographic Map of Italy [51] at a scale of 1:25,000, created in the 1950s. It is the cartographic document that best represents the extension of the urban areas of the time, probably built using the tuff extracted on site. In fact, if we superimpose the spatial distribution of the known cavities on the areas representing the current historical centers of the cities, deduced from the Map of Italy, the overlapping network of cavities and historic city centers is immediately evident (Figure 12a,b).

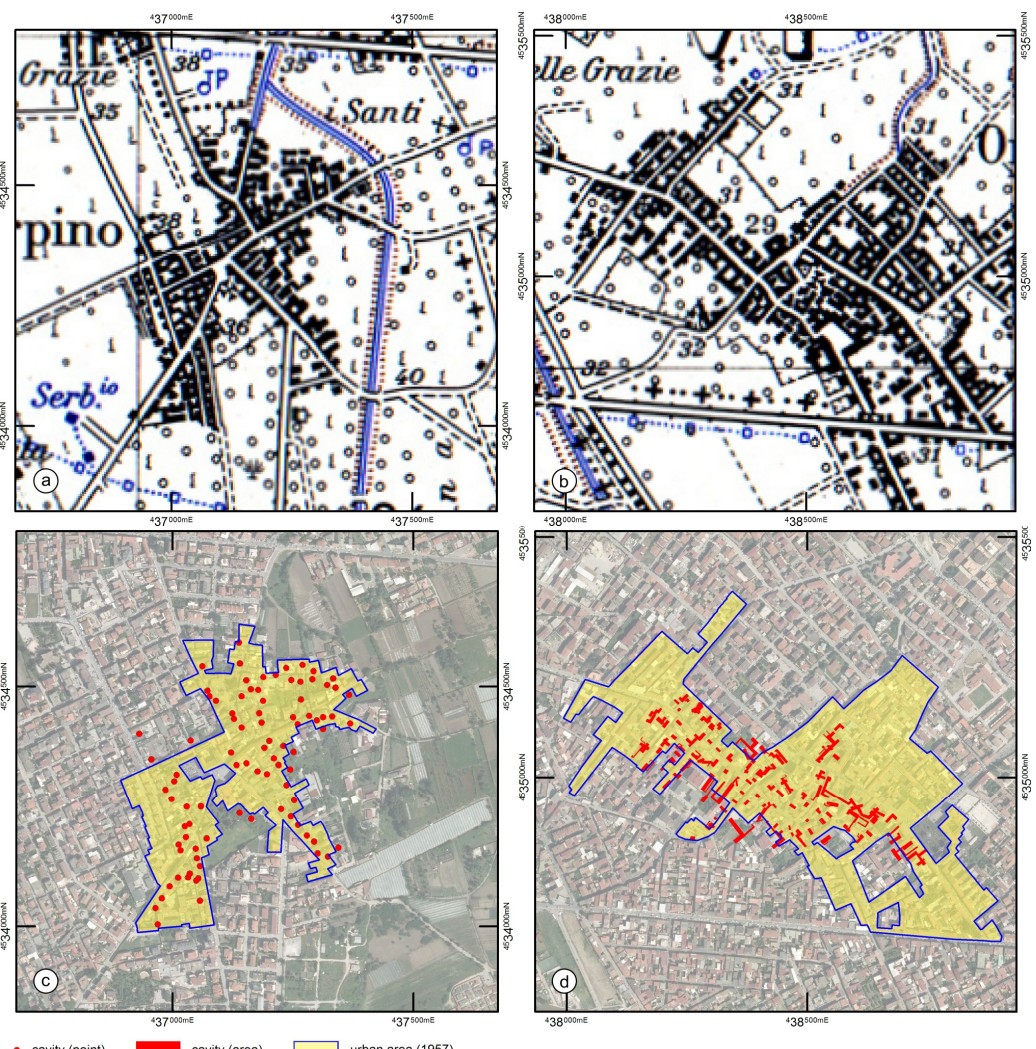

**Figure 12.** Example of distribution of the known cavities in two cities of the studied area (**a**,**b**) and comparison with the extension of the relative historical centers up to 1957 represented in the Topographical Map of Italy of the Military Geographical Institute: (**a–c**) Sant'Arpino; (**b–d**) Orta of Atella.

To give a dimension of the extension of the phenomenon, considering the number of caveal systems known up to now (reported as points, whether they are cavities surveyed or hypothesized on the presence of vertical access, Figure 11), in relation to the areal extension of the historic centers in 1957 (approximately equal to 5.82 km$^2$; cf. Figure 3), the average point density of voids is equal to 458 points/km$^2$.

## 5. Conclusions

The study provides the first database of the geological underground in the western province of Caserta, north of Naples. The database is designed to be managed into a GIS environment in order to provide a useful tool for monitoring and managing the cavities for risk mitigation and tourism enhancement.

The knowledge of the geological and stratigraphic architecture of subsoil in fact allows to hypothesize the presence of cavities and their depth where they are unknown. Further information can derive from the analysis of ancient cartographic documents since the beginning of the XIX century. The latter has revealed a close relationship between the occurrence of cavities in the subsoil and the expansion of the city until the 1970s, when the use of cement completely replaced the extraction of tuff.

Alongside historical and geological data, a database framework was elaborated that includes all the recognized cavities and their architectural characteristics, with a view

to both monitoring for stability purposes and valorization for tourism purposes. In fact, cavities were often buried after excavation without considering their extension in the subsoil and the presence of overlapping urban systems. The occurrence of sinkholes closely linked to the presence of cavities in the city causes social, economic and environmental problems that require timely interventions. The knowledge of such a cavity system is a useful contribution to the hazard evaluation in densely urbanized areas. More recently, moreover, the diffusion of promotional events aimed at opening the subsoil of cities to tourism is opening up new and unprecedented prospects for economic development.

In this view, such a knowledge and mapping of cavities becomes a priority for local authorities to mitigate the risk of collapses, guarantee safety and exploit their potential as a resource for tourism and the economy.

**Author Contributions:** Conceptualization, D.R. and M.V.; methodology, D.R. and M.V.; software, M.V.; validation, D.R., M.V. and M.A.F.; investigation, M.V., M.A.F. and C.B.; data curation, M.V. and M.A.F.; writing—original draft preparation, D.R., M.V., M.A.F. and C.B.; writing—review and editing, D.R.; supervision, D.R. All authors have read and agreed to the published version of the manuscript.

**Funding:** The scholarship and research activities of Maria Assunta Fabozzi were part of the Collaboration Research Program on "Census, analysis and evaluation of the Cavity System" present in the territory of the Hydrographic District of the Southern Apennines. This research was also funded by the SEND intra-university project, funded by the V:ALERE 2020 Program (VAnviteLli pEr la RicErca) of the University of Campania "Luigi Vanvitelli" (Grant ID: B68D19001880005) to DR (WP coordinator).

**Data Availability Statement:** Publicly available datasets were analyzed in this study. The data can be found here: https://www.scopus.com/ (accessed 12 July 2022).

**Conflicts of Interest:** The authors declare no conflict of interest. The funders had no role in the writing of the manuscript.

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
