# Peer review of "Artificial Cavities in the Northern Campania Plain: Architectural Variability and Cataloging Challenge"

_heritage, doi:10.3390/heritage6070289_

Round 1

Reviewer 1 Report

This paper presents “Artificial cavities in the northern Campania Plain: architectural variability and cataloging challenge”.

The topic is very interesting and within the scope of the journal. It is well written. So, the reviewer recommends the paper for publication.

I would like to highlight on the one hand, the contribution this paper does to the underground cavities’ studies, to historical heritage studies, and on the other hand, the use of data analysis methods and techniques.

I recommend the typographical revision of the words written in "local lingo", sometimes they are written in inverted commas, others without nothing.

Author Response

Dear Reviewer,
we kindly acknowledge your insightful comments. We have revised the English of the text and corrected the reported typographical errors.
Further modifications were suggested by Reviewer 2 and the manuscript has been greatly improved.

Kind regards

The Authors

Reviewer 2 Report

Good article with a good synthesis about the management of anthropic cavities.

 In all the article, please change.

Hypogea, Hypogeum: This term is generally used only for graves dug underground, and not as a synonym for the word cavity. Please change Hypogea in Cavity if it is not for a underground grave.

Mining: as your cavities are mainly underground quarries, and not mines, why do not use quarrying or underground quarrying in place of mining ?

Cave : the word cave is sometimes uses (line 245, or 298 for example), replace the word cave by cavity

More comment in the word file linked

Author Response

Dear Reviewer,

we kindly acknowledge your insightful comments. The figures and the manuscript were modified accordingly. Replies to specific comments are listed below.

In all the article, please change:

Hypogea, Hypogeum: This term is generally used only for graves dug underground, and not as a synonym for the word cavity. Please change Hypogea in Cavity if it is not for a underground grave.

Mining: as your cavities are mainly underground quarries, and not mines, why do not use quarrying or underground quarrying in place of mining ?

Cave : the word cave is sometimes uses (line 245, or 298 for example), replace the word cave by cavity

ANSWER: We changed the text accordingly

Line 11: when, please add a datation of the cavities in the abstract

Lines 20-21: monitoring and management of what ? the cavities, the risk, the cultural heritage ?

Line 32: unpredictability -> their relative unpredictability

Line 40 : soils -> subsoils

Line 55-56 : is related to REF 18 ? if so, add ref.

Lines 68-69: underground quarrying not mining

Line 76: the ref 30 is not about underground quarries, please change this ref by another one or change your sentence : replace “extensive underground mining activity” by “extensive quarrying activity”.

ANSWER: We changed the text accordingly

Fig 1 : change Tertiary by Paleogene (if it is Paleogene)

ANSWER: Done

Line 96 : pyroclastic (and not piro) ?

Line 119-120 : add some REF used ?

Line 126 : Previous studies : which ones, add REF

ANSWER: We changed the text accordingly

Line 134-135 : “Population, testing, Maintenance”, please explain or delete this sentence

ANSWER: We have preferred to delete the sentence because it would have taken up too much space to explain these three terms which are in common use in database management. It was beyond the scope of the paper

Line 174 : heterotopic ? Heterometric ?

ANSWER: The correct word is heteropic and it is a commonly used geological term which indicates the lateral passage between different lithologies.

Line 193 : Pipes (explain)

Linee 207 : did they really drill or this word is not appropriate ? Drilling is a very specific practice and not very use to create access or to create a pit.

ANSWER: We changed the text accordingly

Fig 4 : Legend need to be put in a better place in this fig, here it’s too close to the grey layer. (change the word “cave” into “anthropic cavities”.

Fig 5 : Why the cavities don’t go to the bottom limit of the layer 2 ? Can you precise that and how many meters of the layer 2 were not quarried ?

ANSWER: Figures were changed accordingly

Line 226 : pulvis puteolanus (intalic)

ANSWER: Done

Fig 6: add scales

Fig 7: add scales. Incline -> ramp ?

ANSWER: Done

Line 245-247: why this system of tunnels is not of your fig 5 ?

Line 248 : add a subtitle : Uses of cavities ?

Line 262-263: please justify the exceptional aspect according to unesco criteria or delete this sentence

ANSWER: We changed the text accordingly

Fig 8: add scales, and location. Title too unprecise

ANSWER: The purpose of the figure was to collect some photographic examples of ways of using the cavities, regardless of location. In each case we have implemented the explanation and included the location.

Line 274 : few cavities ? How many, please give more detailed information.

Line 276 : once abandoned these cavities may undergo degradation ? why degradation only undergo after the abandon ? In theory the degradation begin after the quarrying, not only the abandon of the cavitiy – don't confuse the beginning of degradation with abandon, which leads to a lack of maintenance.

Fig 9: some scales needed

Line 286 4.3. not already page 9 or it is different ? Please precise this

Line 295: add Ref (18 or other Ref).

ANSWER: Done

Line 290-291 : no information about the age or the datation of the cavity in the database ?

ANSWER: This information is included in the cataloging scheme of the National Catalog of Artificial Cavities, to which the tables elaborated in this manuscript will link.

Fig 10 : the text is too little when print.

ANSWER: The font size was changed

Line 312-313 : the dates are really too imprecise, isn't it possible to improve this historical aspect in the article? Without dating, it's no longer heritage, it's risk/...

ANSWER: It is possible to improve the historical aspect place by place, in relation to the age of the individual buildings that subtend cavities. We have included a comment in the text.

Fig 11: add scales

ANSWER: Done

Lines 335-341, and fig 12 It's clear to me that your 1950s map isn't old enough, and that your correlation would be much better if you used older maps. Please add the objective in your conclusion.

ANSWER: The oldest maps provide partial evidence of the extension of the historic centers in which, until the early 1960s, people built using the tuff extracted from the subsoil. From that period onwards, construction techniques were based solely on the use of cement and tuff extraction was abandoned. For this reason, the maps of the 1950s are the most significant for having an idea of the maximum extension of the borders built with tuff and, consequently, of the possible presence of underground cavities.

Fig 12: put the old map at the top of the fig, and youy GIS data at the bottom (= reverse the position of the maps).

ANSWER: Done

Line 359-360 : informations from XIX century maps ? Yes !!! So why you simply used maps from 1950… use XIX century maps (and olders if you can). Please add this an objective for the future in your conclusion.

ANSWER: See previous comment